# IMPROVING LoRA WITH VARIATIONAL LEARNING

## ABSTRACT

Bayesian methods have recently been used to improve calibration of LoRA fine-tuning but there is still room for improvements. For instance, with Laplace's method no effective gains in accuracy are seen while variational learning can sometimes even harm it and increase both runtime and implementation complexity. Here, we propose two simple modifications to variational learning that fix all of these issues. First, we reduce cost and simplify implementation by adapting the recently proposed IVON optimizer for LoRA training. Second, we propose new scaling and pruning techniques for posteriors to improve the accuracy-uncertainty trade-off. Empirically these modifications consistently yield multiple benefits over Adam where (a) both accuracy and calibration are boosted; (b) accuracy improves with longer training while overfitting is reduced; (c) test-time scaling is boosted for generation tasks; and (d) data efficiency during training is also improved. Our work proposes new modifications to variational learning that improve many aspects of the standard LoRA training.

## 1 INTRODUCTION

Low-Rank Adaptation (LoRA) (Hu et al., 2022) is a widely used parameter-efficient finetuning technique for large-scale pretrained models. By introducing small, low-rank adapters, it enables billion-scale Large Language Models (LLMs) to be finetuned on a single consumer-grade GPU, making it the go-to method for finetuning LLMs with limited compute. Since its introduction, many new improvements have been proposed in various directions, such as parameter efficiency (He et al., 2022; Ding et al., 2023; Zhang et al., 2023a; Kopiczko et al., 2024), performance under quantization (Dettmers et al., 2024; Xu et al., 2024; Li et al., 2024), and rank adaptation (Zhang et al., 2023b; Ding et al., 2023; Valipour et al., 2023; Lialin et al., 2024).

Recent works have also explored Bayesian methods to improve the generalization and calibration of LoRA finetuning. While they work well to some extent, there is still room for improvement. Laplace-LoRA (Yang et al., 2024) estimates a posterior by using Laplace's method on trained LoRA parameters, but this requires additional post-hoc changes, including calculating a Kronecker-factored Hessian and model linearization. Despite increasing the overhead, only marginal improvements in accuracy are obtained, mainly because the posterior mean is learned using non-Bayesian methods. BLoB (Wang et al., 2024) improves this by jointly learning the posterior mean and covariance with Bayes by Backprop (Blundell et al., 2015), a popular variational learning method, yet posterior sampling hurts accuracy. BLoB also requires additional computational cost and implementation tricks, such as flipout (Wen et al., 2018) which introduces nontrivial changes to LoRA's forward pass logic and can conflict with other customizations of LoRA. Ideally we would like a simpler alternative that can bring more benefits with less overhead.

In this paper, we propose simple modifications to variational learning to fix the above issues and improve LoRA finetuning of LLMs. Specifically, we use a recent natural-gradient variational learning algorithm called Improved Variational Online Newton (IVON) (Shen et al., 2024), for its simplicity and efficiency. We propose practical adaptations of IVON to better suit LoRA finetuning, and introduce new scaling and pruning techniques for the learned posterior to explicitly manage the accuracy-uncertainty trade-off at test time, which existing methods do not consider. The resulting method, IVON-LoRA, requires minimal changes to the training pipeline and has minimal overhead over Adam, yet effectively learns a good posterior over LoRA parameters.

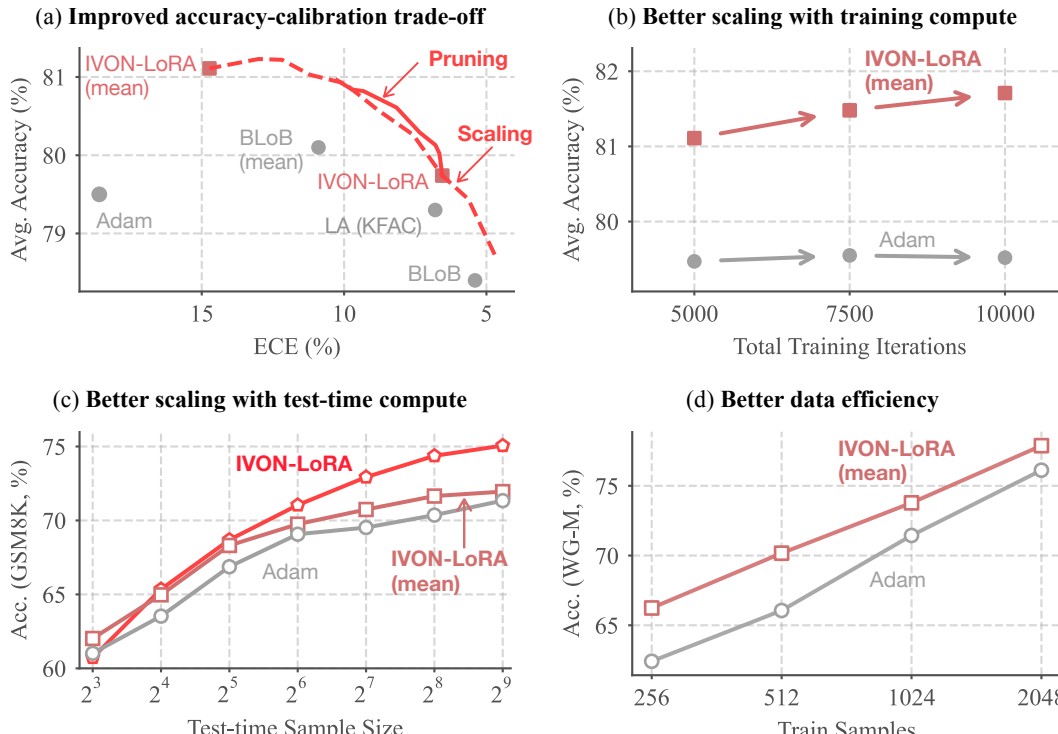

Figure 1: We improve traditional LoRA finetuning by proposing new modifications for variational learning. (a) Our new method IVON-LoRA can obtain the best accuracy-calibration trade-off by using posterior scaling and pruning techniques, and outperforms other Bayesian approaches such as BLoB and Laplace-LoRA. (b) IVON-LoRA scales better with more training computation compared to Adam. (c) IVON-LoRA scales better with more test-time computation by combining the learned posterior with uncertainty-aware MBR decoding (Daheim et al., 2025). (d) IVON-LoRA has better data efficiency than Adam, achieving similar accuracy with less training data.

IVON-LoRA consistently improves generalization and calibration of LoRA finetuning across datasets, outperforming Adam and Bayesian baselines such as Laplace-LoRA and BLoB. Notably, IVON-LoRA on average improves accuracy by 1.6% and reduces ECE by 3.9% over Adam for fine-tuning Llama-3.2-3B on a set of commonsense reasoning tasks used by existing works (Yang et al., 2024; Wang et al., 2024), and supports a flexible trade-off between accuracy and calibration by adjusting the posterior temperature or pruning ratio at test time. Besides, it scales better with increased computation both during training and at test time, and has better data efficiency, which is increasingly important as compute grows faster than curated datasets. Overall, our work proposes new modifications to variational learning that improve many aspects of the LoRA finetuning of LLMs.

## 2 BACKGROUND

### 2.1 LOW-RANK ADAPTATION (LoRA)

The large sizes of current LLMs make it hard for many practitioners to finetune the full model due to resource constraints. To tackle this, Low-Rank Adaptation (LoRA) Hu et al. (2022) inserts new parameters as a low-rank decomposition of the update applied to the original parameters during finetuning. These new parameters can then be merged with the original parameters to not increase inference overhead. Formally, given a weight matrix $\mathbf{W}_0 \in \mathbb{R}^{d \times k}$ of a pretrained model, LoRA introduces a low-rank decomposition

$$\Delta \mathbf{W}_0 = \mathbf{BA},$$

where $\mathbf{A} \in \mathbb{R}^{r \times k}$, $\mathbf{B} \in \mathbb{R}^{d \times r}$, and $r \ll \min(d, k)$. The new weight matrix $\mathbf{W}$ is then:

$$\mathbf{W} = \mathbf{W}_0 + \Delta \mathbf{W}_0 = \mathbf{W}_0 + \mathbf{BA}.$$

The low-rank matrices $\mathbf{A}$ and $\mathbf{B}$ are initialized such that $\Delta\mathbf{W}_0 = \mathbf{B}\mathbf{A} = 0$ to preserve the original $\mathbf{W}_0$ at the beginning of training. During training, only $\mathbf{A}$ and $\mathbf{B}$ are optimized, while $\mathbf{W}_0$ remains frozen. This drastically reduces the number of trainable parameters from $O(dk)$ to $O(2dr)$. For linear layers, the forward pass with LoRA can be computed as:

$$\boldsymbol{h} = \mathbf{W}_0\boldsymbol{x} + \Delta\mathbf{W}_0\boldsymbol{x} = \mathbf{W}_0\boldsymbol{x} + \mathbf{B}\mathbf{A}\boldsymbol{x},$$

where $\boldsymbol{x} \in \mathbb{R}^k$ is the input and $\boldsymbol{h} \in \mathbb{R}^d$ the output. For Transformers (Vaswani et al., 2017), which are our main focus, LoRA is typically applied to the query and value matrices of the attention layers.

## 2.2 BAYESIAN LoRA

Despite its computational efficiency, LoRA can lead to overconfident models (Wang et al., 2023) and lag behind full finetuning in terms of accuracy (Biderman et al., 2024). To tackle this, Bayesian variants of LoRA have recently been proposed. Here we briefly review two representative methods: Laplace-LoRA (Yang et al., 2024) and BLoB (Wang et al., 2024).

### 2.2.1 LAPLACE-LoRA

After ordinary LoRA finetuning, Laplace-LoRA (Yang et al., 2024) treat the learned LoRA weight $\mathbf{m}$ (concatenation of flattened $\mathbf{A}$ and $\mathbf{B}$ matrices) as a MAP estimate, and uses Laplace's method (MacKay, 1992) to estimate a Gaussian posterior distribution over $\mathbf{m}$. With an isotropic Gaussian prior $p(\boldsymbol{\theta}) = \mathcal{N}(\boldsymbol{\theta} \mid 0, \mathbf{I}/s_0)$, it gives the following posterior:

$$q(\boldsymbol{\theta}) = \mathcal{N}(\boldsymbol{\theta} \mid \mathbf{m}, (\mathbf{F} + s_0\mathbf{I})^{-1}),$$

where $\mathbf{F}$ is the Kronecker-factored (KFAC) approximation (Martens & Grosse, 2015; Ritter et al., 2018) of the Fisher information matrix computed at $\mathbf{m}$. Laplace-LoRA further linearizes the model (Immer et al., 2021), which gives the posterior on the model output logits:

$$f_{\boldsymbol{\theta}}(\mathbf{x}) \sim \mathcal{N}(f_{\mathbf{m}}(\mathbf{x}), \nabla_{\boldsymbol{\theta}} f_{\boldsymbol{\theta}}(\mathbf{x})|_{\mathbf{m}}^{\top} (\mathbf{F} + s_0\mathbf{I})^{-1} \nabla_{\boldsymbol{\theta}} f_{\boldsymbol{\theta}}(\mathbf{x})|_{\mathbf{m}}),$$

where $f_{\boldsymbol{\theta}}(\mathbf{x})$ is the model output for input $\mathbf{x}$ given LoRA parameters $\boldsymbol{\theta}$. Then predictions can be made by averaging samples from this posterior.

There are several problems with Laplace-LoRA though. First, it only estimates the posterior covariance and the mean is obtained by standard (non-Bayesian) training, therefore no or marginal gains in accuracy are observed. Second, it requires an additional pass through the training data to compute $\mathbf{F}$ which can be expensive for large datasets (sometimes training data is not even available after training). Finally, model linearization requires the Jacobian $\nabla_{\boldsymbol{\theta}} f_{\boldsymbol{\theta}}(\mathbf{x})$ for the neural network outputs $f_{\boldsymbol{\theta}}(\mathbf{x})$. This can be prohibitive for LLMs in a standard setting for next-token prediction due to requiring storage of $\mathcal{O}(d|\mathcal{V}|)$ for an output vocabulary $\mathcal{V}$ and number of parameters $d$.

### 2.2.2 BLoB

A recently proposed method, BLoB (Wang et al., 2024), circumvents some of these issues by directly learning the mean $\mathbf{m}$ and diagonal covariance $\mathbf{v}$ during training with Bayes by Backprop (Blundell et al., 2015). It further introduces tricks, such as an asymmetric Bayesianization that only treats $\mathbf{A}$ probabilistically while keeping $\mathbf{B}$ deterministic, a new covariance parameterization $\mathbf{v} = \mathbf{u}^2$ to ensure positivity and promote fast convergence, and a new variant of flipout (Wen et al., 2018) to increase sampling efficiency.

While BLoB can improve both accuracy and calibration over Adam, it has a suboptimal trade-off between accuracy and calibration. Good accuracy is observed when just using the mean of the posterior but calibration improvement is limited, On the other hand, sampling further improves calibration but degrades accuracy. Using only the posterior mean at test time gives better accuracy but calibration improvement is limited, while posterior sampling further improves calibration but hurts accuracy. Also, BLoB introduces additional computational overhead and implementation complexity. It requires an extra optimizer for $\mathbf{u}$ and a few new hyperparameters to tune, as well as modifications to the LoRA implementation and training loop which can conflict with other customizations.

## 2.3 Improved Variational Online Newton (IVON)

IVON (Shen et al., 2024) is a recent variational learning method that optimizes a variational-Bayesian objective similar to Bayes by Backprop (Blundell et al., 2015). Formally, instead of minimizing the standard training loss $\ell(\boldsymbol{\theta})$ as in Adam, IVON minimizes the following objective where an expectation of $\ell(\boldsymbol{\theta})$ over a posterior distribution $q(\boldsymbol{\theta})$ is used:

$$\min_{q(\boldsymbol{\theta})} \ \mathbb{E}_{q(\boldsymbol{\theta})}\left[\ell(\boldsymbol{\theta})\right] + \lambda^{-1} \, \mathbb{D}_{\mathrm{KL}}[q(\boldsymbol{\theta}) \, \| \, p(\boldsymbol{\theta})], \tag{1}$$

where $\lambda$ is a scaling factor. Specifically, IVON uses a diagonal Gaussian posterior $q(\boldsymbol{\theta}) = \mathcal{N}(\mathbf{m}, \mathrm{diag}(\mathbf{v}))$ with a zero mean isotropic Gaussian prior $p(\boldsymbol{\theta}) = \mathcal{N}(\boldsymbol{\theta} \mid 0, \mathbf{I}/\delta)$, where $\delta$ is the weight decay.

The main advantage of IVON is that it only requires a few lines of training code to be changed and has minimal overhead, thanks to the nearly identical implementation of IVON and Adam. The key point is that estimation of $\mathbf{v}$ is done automatically through the scale vector $\mathbf{h}$ that adapts the learning rate. Specifically, the variance is set as $\mathbf{v} = 1/(\lambda(\mathbf{h} + \delta))$ where $\mathbf{h} = \nabla^2 \ell(\boldsymbol{\theta})$ is the diagonal Hessian. Therefore, $\mathbf{v}$ can be obtained for free by estimating gradients at a perturbed $\boldsymbol{\theta} \sim \mathcal{N}(\mathbf{m}, \mathrm{diag}(\mathbf{v}))$ to estimate the expectation in Eq. 1 and using the reparametrization trick to get $\mathbf{h}$. In practice, even using one Monte-Carlo sample $\boldsymbol{\theta} \sim \mathcal{N}(\mathbf{m}, \mathrm{diag}(\mathbf{v}))$ performs well and incurs almost no overhead (see Sec. 4.3 for a benchmark). For further details of IVON, we refer to Shen et al. (2024). Overall, IVON is an easy-to-use alternative to existing Bayesian approaches that require additional overheads due to post-processing, additional passes through the data, and cumbersome implementation changes.

## 3 Adapting IVON for LoRA Finetuning

In this section, we discuss a few adaptations of IVON to improve it for LoRA finetuning of LLMs. We provide the pseudocode of LoRA finetuning with IVON-LoRA in Alg. 1, with the modifications discussed in Sec. 3.1 and Sec. 3.2 highlighted.

### 3.1 Finding Good Values for the Prior

In the original design of IVON, the weight decay $\delta$ is used as the prior precision. However, there is no good way to set $\delta$ for either LoRA finetuning or IVON, other than tuning it as a hyperparameter. Also, the low-rank matrices $\mathbf{A}$ and $\mathbf{B}$ are heterogeneous (due to their different initialization and positions in the network), and using the same prior for all $\mathbf{A}$ and $\mathbf{B}$ matrices can be suboptimal.

To tackle these problems without introducing additional hyperparameters, we instead assign a separate prior to each $\mathbf{A}$ and $\mathbf{B}$ matrix by setting it to the value that minimizes the KL divergence term in Eq. 1 for that matrix for each training step. Specifically, the optimal prior precision for the $i$-th LoRA matrix with posterior $q(\boldsymbol{\theta}) = \mathcal{N}(\mathbf{m}_i, \mathrm{diag}(\mathbf{v}_i))$ is given by the following closed-form expression (Graves, 2011, Eq. 15):

$$\delta_i = \left( \frac{1}{d_i} \sum_{j=1}^{d_i} (m_{ij}^2 + v_{ij}) \right)^{-1}, \tag{2}$$

where $d_i$ is the number of parameters in that matrix and $m_{ij}$ and $v_{ij}$ are the $j$-th entries of $\mathbf{m}_i$ and $\mathbf{v}_i$, respectively (see App. B for a derivation). Besides, we disable the explicit weight decay in the parameter update step, as a zero weight decay empirically works well for LoRA finetuning. By doing so, we effectively find a good value of the prior precision for each LoRA matrix without introducing new hyperparameters. Note that this only involves few element-wise operations and the overhead is negligible.

### 3.2 The Choice of $\lambda$

$\lambda$, which scales the KL divergence term in Eq. 1, is another important hyperparameter to tune. It can be seen as an effective training data size, where $\lambda = N$ targets a generalized posterior for

$N$ data points (Zellner, 1988). However, simply setting $\lambda$ to $N$ can be suboptimal when $N$ is extremely small, as it scales the posterior variance $\mathbf{v}$ to be very large and can lead to unstable training. Though Shen et al. (2024) suggest to set $\lambda > N$ on finetuning tasks with a small dataset, a solid rule of thumb for choosing $\lambda$ is still missing.

Qualitatively, $\lambda$ should increase with the number of samples $N$ to downweight the KL term, but a linear scaling can fail when $N$ is too small. Considering this, we instead use a heuristic to set $\lambda = \gamma \sqrt{N}$ where $\gamma$ is a scaling factor to be tuned. We find this heuristic works well across datasets of different sizes, and allows us to use a smaller $\beta_2$ than the value suggested in Shen et al. (2024), which promotes faster learning of $\mathbf{v}$.

### 3.3 Test-time Posterior Scaling and Pruning

One problem with existing Bayesian adaptation of LoRA such as BLoB, is the imbalanced improvement of generalization and calibration. For example, BLoB improves test accuracy, but the improvement in calibration is limited when testing with the posterior mean. On the other hand, averaging predictions of models sampled from the posterior further improves calibration but accuracy degrades severely. Ideally, we would like a more graceful trade-off between accuracy and calibration which can be adjusted based on the application.

In this work, we propose to achieve such trade-off by either adjusting the temperature of the posterior or pruning parameters at test-time. For posterior temperature, we simply use a scaled $\lambda_{\text{test}} = \tau\lambda$ at test time, where $\tau > 0$ is a scaling factor. Then the posterior variance at test time becomes $\mathbf{v}_{\text{test}} = 1/(\tau\lambda(\mathbf{h} + \delta))$. Setting $\tau = 1$ recovers the standard posterior to sample from, while $\tau \to \infty$ recovers the point estimate at posterior mean $\mathbf{m}$. By adjusting $\tau$, we can smoothly scale the posterior variance to trade-off between the accuracy and calibration.

Surprisingly, we find that pruning a small fraction of parameters at test time can achieve an even better trade-off. Here we use similar metrics as in LeCun et al. (1989); Graves (2011); Chen & Garner (2024) and prune parameters with the smallest signal-to-noise ratio $|m_j|/\sqrt{v_j}$, where $m_j$ and $v_j$ are the $j$-th entries of $\mathbf{m}$ and $\mathbf{v}$, respectively. At test time, we set the pruned parameters to 0 and also set their variance to 0 to avoid sampling them. We empirically find such pruning similarly improves accuracy at the cost of calibration, but improves accuracy more at the same cost under a mild pruning ratio.

## 4 Experiments

In this section, we evaluate our method on established benchmarks. In Sec. 4.1, we compare IVON-LoRA with baselines on commonsense reasoning tasks. In Sec. 4.2, we show that variational learning with IVON-LoRA scales better with more training and test-time computation, and has better data efficiency. We provide additional results on the GLUE benchmark in App. C.2.

### 4.1 IVON-LoRA Improves Generalization and Calibration

Following the settings in Yang et al. (2024) and Wang et al. (2024), we evaluate our method on commonsense reasoning tasks. We finetune Llama-3.2-3B (Grattafiori et al., 2024) on six commonsense reasoning datasets, including WinoGrande-S (WG-S), WinoGrande-M (WG-M) (Sakaguchi et al., 2021), ARC-Challenge (ARC-C), ARC-Easy (ARC-E) (Clark et al., 2018), OpenBookQA (OBQA) (Mihaylov et al., 2018), and BoolQ (Clark et al., 2019). We measure accuracy and Expected Calibration Error (ECE) on the validation splits and use Negative Log-Likelihood (NLL) as an additional metric for calibration, since ECE may be unreliable when annotators disagree (Baan et al., 2022). We compare our results to baselines including standard LoRA finetuning with Adam, Laplace-LoRA (Yang et al., 2024) and BLoB (Wang et al., 2024). For BLoB and IVON-LoRA, we report the results acquired by either using the mean of the posterior $\mathbf{m}$ (indicated by the suffix "@mean") or by using an averaged prediction over 10 samples from the posterior. Please refer to App. D.2 for details on the experimental setup.

Results are shown in Tab. 1 and Tab. 2. As shown in Tab. 1, IVON-LoRA@mean outperforms baseline methods on most datasets in terms of accuracy, often by a large margin. IVON-LoRA@mean on average improves accuracy by 1.6% over Adam and 1.0% over BLoB@mean. Our method also

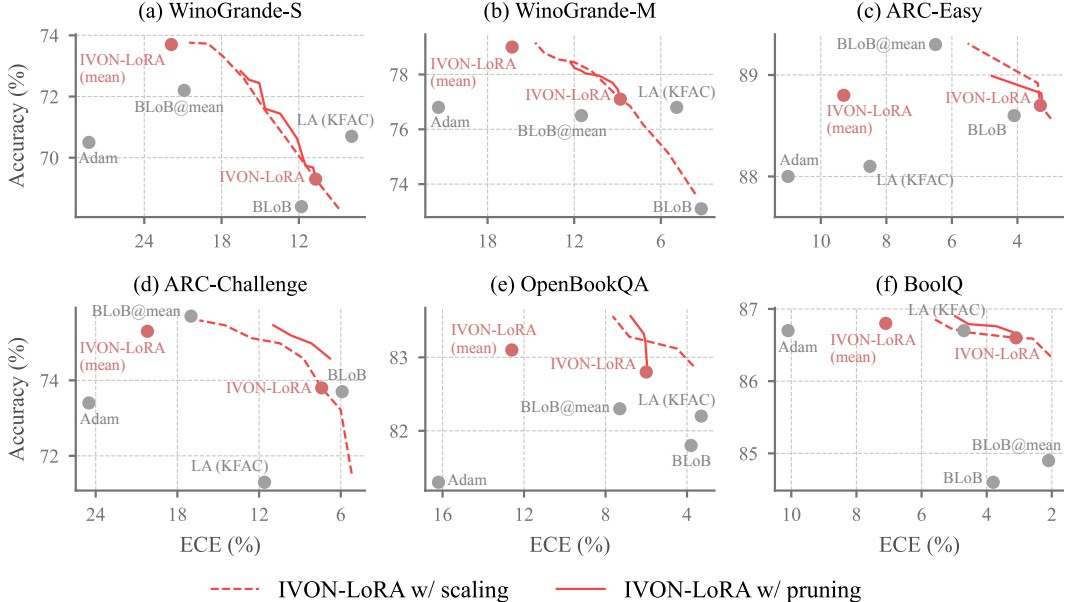

Figure 2: Trade-off between accuracy and calibration can be achieved by adjusting the posterior temperature or pruning ratio at test time. Adjusting posterior temperature gives a more flexible trade-off, while pruning performs similar or better under a certain range with mild pruning ratio. We show results on six commonsense reasoning datasets with Llama-3.2-3B. The red curves indicate the Pareto-optimal frontiers over posterior temperature and pruning ratio, respectively.

Table 1: Being easy-to-use and overhead-free, IVON-LoRA significantly improves the generalization of LoRA finetuning (indicated by the validation accuracy) and enhances calibration. In later section we will demonstrate that our method enables smooth trade-off between accuracy and calibration and generally performs better than baselines. Here we compare methods on LoRA finetuning of the Llama-3.2-3B model across commonsense reasoning datasets, with subscripts indicating standard error of the mean across 5 runs. We show the relative improvements over Adam in blue.

| Metrics | Methods | WG-S | ARC-C | ARC-E | WG-M | OBQA | BoolQ | Avg. |
|---|---|---|---|---|---|---|---|---|
| **ACC ↑** | Adam | $70.5_{0.2}$ | $73.4_{0.7}$ | $88.0_{0.3}$ | $76.8_{0.3}$ | $81.3_{0.6}$ | $86.7_{0.1}$ | 79.5 |
| | BLoB@mean | $72.2_{0.2}$ | $75.7_{0.7}$ | $89.3_{0.2}$ | $76.5_{0.4}$ | $82.3_{0.3}$ | $84.9_{0.2}$ | (+0.6) 80.1 |
| | IVON-LoRA@mean | $73.7_{0.2}$ | $75.3_{0.3}$ | $88.8_{0.4}$ | $79.0_{0.4}$ | $83.1_{0.4}$ | $86.8_{0.1}$ | (+1.6) 81.1 |
| **ECE** ($\times 100$) ↓ | Adam | $28.3_{0.2}$ | $24.5_{0.7}$ | $11.0_{0.4}$ | $21.4_{0.3}$ | $16.2_{0.6}$ | $10.1_{0.2}$ | 18.6 |
| | BLoB@mean | $20.9_{0.3}$ | $17.0_{0.8}$ | $6.5_{0.1}$ | $11.5_{0.4}$ | $7.3_{0.3}$ | $2.1_{0.3}$ | (-7.7) 10.9 |
| | IVON-LoRA@mean | $21.9_{0.2}$ | $20.2_{0.3}$ | $9.3_{0.4}$ | $16.3_{0.3}$ | $12.6_{0.2}$ | $7.1_{0.1}$ | (-3.9) 14.7 |
| **NLL ↓** | Adam | $3.14_{0.10}$ | $2.50_{0.10}$ | $1.01_{0.08}$ | $1.75_{0.08}$ | $1.29_{0.06}$ | $0.57_{0.01}$ | 1.71 |
| | BLoB@mean | $1.02_{0.01}$ | $1.00_{0.02}$ | $0.39_{0.01}$ | $0.59_{0.01}$ | $0.53_{0.01}$ | $0.35_{0.01}$ | (-1.06) 0.65 |
| | IVON-LoRA@mean | $1.47_{0.03}$ | $1.41_{0.02}$ | $0.66_{0.02}$ | $0.89_{0.02}$ | $0.79_{0.01}$ | $0.38_{0.00}$ | (-0.78) 0.93 |
| **Brier ↓** | Adam | $0.57_{0.00}$ | $0.52_{0.01}$ | $0.22_{0.01}$ | $0.44_{0.00}$ | $0.34_{0.00}$ | $0.23_{0.00}$ | 0.39 |
| | BLoB@mean | $0.47_{0.00}$ | $0.40_{0.01}$ | $0.18_{0.00}$ | $0.35_{0.01}$ | $0.26_{0.00}$ | $0.22_{0.00}$ | (-0.08) 0.31 |
| | IVON-LoRA@mean | $0.46_{0.01}$ | $0.43_{0.01}$ | $0.17_{0.00}$ | $0.36_{0.01}$ | $0.27_{0.01}$ | $0.20_{0.00}$ | (-0.08) 0.31 |

exhibits visibly improved calibration compared to Adam baseline. Notably, these improvements are achieved with no test-time overhead at all, since only the mean of the posterior is used for prediction.

Next, as shown in Tab. 2, IVON-LoRA maintains the superiority in a setting that uses the posterior to make predictions (posterior sampling for BLoB and IVON-LoRA, and model linearization for Laplace-LoRA). IVON-LoRA still achieves a 0.2% accuracy gain over Adam, while both BLoB and Laplace-LoRA *degrade* accuracy. The calibration of IVON-LoRA is comparable to BLoB and Laplace-LoRA, and significantly outperforms Adam baseline. Notably, this is achieved with neither

Table 2: In a more Bayesian setting, IVON-LoRA significantly improves calibration and still maintain a decent accuracy, while both LA and BLoB suffer from accuracy degradation. The experimental setup is similar to Tab. 1, except that methods use a Bayesian approach at test time (model linearization for LA, posterior sampling for BLoB and IVON-LoRA). Subscripts indicating standard error of the mean across 5 runs. We show the relative metric changes over Adam in parentheses, with improvements in blue and degradation in red.

| Metrics | Methods | WG-S | ARC-C | ARC-E | WG-M | OBQA | BoolQ | Avg. |
|---|---|---|---|---|---|---|---|---|
| ACC ↑ | Adam | $70.5_{0.2}$ | $73.4_{0.7}$ | $88.0_{0.3}$ | $76.8_{0.3}$ | $81.3_{0.6}$ | $86.7_{0.1}$ | 79.5 |
| | + LA (KFAC) | $70.7_{0.3}$ | $71.3_{1.2}$ | $88.1_{0.2}$ | $76.8_{0.2}$ | $82.2_{0.2}$ | $86.7_{0.2}$ | (-0.2) 79.3 |
| | + LA (diag) | $70.4_{0.5}$ | $61.8_{0.3}$ | $80.5_{0.3}$ | $77.5_{0.4}$ | $81.4_{0.3}$ | $86.7_{0.2}$ | (-3.1) 76.4 |
| | BLoB | $68.4_{0.5}$ | $73.7_{0.5}$ | $88.6_{0.3}$ | $73.1_{0.3}$ | $81.8_{0.3}$ | $84.6_{0.2}$ | (-1.1) 78.4 |
| | IVON-LoRA | $69.3_{0.4}$ | $73.8_{1.1}$ | $88.7_{0.2}$ | $77.1_{0.4}$ | $82.8_{0.5}$ | $86.6_{0.1}$ | (+0.2) 79.7 |
| ECE ($\times 100$) ↓ | Adam | $28.3_{0.2}$ | $24.5_{0.7}$ | $11.0_{0.4}$ | $21.4_{0.3}$ | $16.2_{0.6}$ | $10.1_{0.2}$ | 18.6 |
| | + LA (KFAC) | $7.9_{0.6}$ | $11.6_{0.9}$ | $8.5_{1.6}$ | $4.9_{0.4}$ | $3.3_{0.2}$ | $4.7_{0.2}$ | (-11.8) 6.8 |
| | + LA (diag) | $17.1_{0.7}$ | $18.9_{0.5}$ | $34.5_{0.5}$ | $14.8_{0.6}$ | $17.1_{1.0}$ | $17.1_{0.4}$ | (+1.3) 19.9 |
| | BLoB | $11.8_{0.4}$ | $5.9_{0.8}$ | $4.1_{0.2}$ | $3.2_{0.1}$ | $3.8_{0.5}$ | $3.8_{0.2}$ | (-13.2) 5.4 |
| | IVON-LoRA | $10.7_{0.3}$ | $7.4_{0.4}$ | $3.3_{0.3}$ | $8.8_{0.3}$ | $6.0_{0.3}$ | $3.1_{0.1}$ | (-12.0) 6.6 |
| NLL ↓ | Adam | $3.14_{0.10}$ | $2.50_{0.10}$ | $1.01_{0.08}$ | $1.75_{0.08}$ | $1.29_{0.06}$ | $0.57_{0.01}$ | 1.71 |
| | + LA (KFAC) | $0.59_{0.00}$ | $0.83_{0.01}$ | $0.42_{0.02}$ | $0.52_{0.01}$ | $0.55_{0.00}$ | $0.37_{0.00}$ | (-1.16) 0.55 |
| | + LA (diag) | $0.66_{0.00}$ | $1.02_{0.01}$ | $0.85_{0.01}$ | $0.55_{0.00}$ | $0.61_{0.01}$ | $0.43_{0.01}$ | (-1.02) 0.69 |
| | BLoB | $0.67_{0.00}$ | $0.73_{0.01}$ | $0.34_{0.01}$ | $0.55_{0.01}$ | $0.50_{0.00}$ | $0.36_{0.01}$ | (-1.19) 0.52 |
| | IVON-LoRA | $0.67_{0.00}$ | $0.80_{0.01}$ | $0.38_{0.01}$ | $0.57_{0.00}$ | $0.53_{0.01}$ | $0.32_{0.00}$ | (-1.17) 0.54 |
| Brier ↓ | Adam | $0.57_{0.00}$ | $0.52_{0.01}$ | $0.22_{0.01}$ | $0.44_{0.00}$ | $0.34_{0.00}$ | $0.23_{0.00}$ | 0.39 |
| | + LA (KFAC) | $0.40_{0.00}$ | $0.42_{0.01}$ | $0.22_{0.01}$ | $0.33_{0.00}$ | $0.26_{0.00}$ | $0.20_{0.00}$ | (-0.08) 0.31 |
| | + LA (diag) | $0.46_{0.00}$ | $0.56_{0.02}$ | $0.45_{0.01}$ | $0.37_{0.01}$ | $0.33_{0.01}$ | $0.26_{0.00}$ | (+0.02) 0.41 |
| | BLoB | $0.43_{0.00}$ | $0.37_{0.01}$ | $0.17_{0.00}$ | $0.36_{0.00}$ | $0.26_{0.00}$ | $0.23_{0.00}$ | (-0.09) 0.30 |
| | IVON-LoRA | $0.42_{0.00}$ | $0.39_{0.01}$ | $0.17_{0.00}$ | $0.34_{0.01}$ | $0.25_{0.01}$ | $0.19_{0.00}$ | (-0.10) 0.29 |

an 1.1% drop in accuracy (as in BLoB) nor a more representative KFAC Hessian or an additional pass through the data to compute that Hessian (as in Laplace-LoRA).

The results in Tab. 1 and Tab. 2 also reveal the difficulty of balancing accuracy and calibration. For both BLoB and IVON-LoRA, sampling from the posterior significantly improves calibration but hurts accuracy. In Fig. 1a and Fig. 2, we show that by scaling the posterior temperature or pruning ratio at test time, we can smoothly trade off between accuracy and calibration. Adjusting posterior temperature gives a more flexible trade-off, indicated by the wide range of ECE and accuracy values achieved by the dashed curves. Pruning performs better or similar on most datasets.

## 4.2 VARIATIONAL LEARNING SCALES BETTER WITH INCREASED COMPUTE

In this section, we evaluate the performance of variational learning on LoRA finetuning under settings where computational resources are richer than data. This reflects the trend in real-world that computational power grows much faster than data. We demonstrate that variational learning benefits more from increased computation. We only consider IVON-LoRA in this section as it generally performs better than BLoB and Laplace-LoRA in Sec. 4.1, but similar trends could also hold for other variational learning methods. We leave the evaluation of them to future work.

**Increasing Training Iterations.** We evaluate the performance of IVON-LoRA and Adam when training for more iterations. We use the same setup as in Sec. 4.1, but train for 10,000 iterations instead of 5,000. We show the results in Tab. 3. While Adam's test accuracy is marginally improved (less than 0.1% on average), the accuracy of IVON-LoRA is further improved by 0.6% on average.

**Increasing Test-Time Computation.** Next, we evaluate the performance of IVON-LoRA when increasing test-time computation. For a more practical setting, we apply IVON-LoRA to reasoning task, where we finetune Qwen-2.5-3B (Team, 2024) on the GSM8k benchmark for math word problem solving (Cobbe et al., 2021). To better utilize the posterior, we use sequence-level uncertainty-aware Minimum Bayes Risk (MBR) decoding (Daheim et al., 2025, Eq. 9). That is, we first sample

Table 3: With longer training, IVON-LoRA further improves the performance over Adam. The experimental setup is the same as in Tab. 1, except that we double the number of training iterations for Adam and IVON-LoRA to 10,000. While Adam does not benefit from more training iterations, IVON-LoRA's average accuracy is further improved by 0.6%. Here we only list accuracy due to space limit, and the full results including ECE and NLL can be found in App. C.1.

| Metrics | Methods | WG-S | ARC-C | ARC-E | WG-M | OBQA | BoolQ | Avg. |
|---------|---------|------|-------|-------|------|------|-------|------|
| | Adam | $70.5_{0.2}$ | $73.4_{0.7}$ | $88.0_{0.3}$ | $76.8_{0.3}$ | $81.3_{0.6}$ | $86.7_{0.1}$ | 79.5 |
| ACC ↑ | Adam (2x iteration) | $71.5_{0.5}$ | $72.6_{0.3}$ | $87.7_{0.4}$ | $77.0_{0.3}$ | $81.8_{0.5}$ | $86.5_{0.1}$ | (+0) 79.5 |
| | IVON-LoRA@mean | $73.7_{0.2}$ | $75.3_{0.3}$ | $88.8_{0.4}$ | $79.0_{0.4}$ | $83.1_{0.4}$ | $86.8_{0.1}$ | (+1.6) 81.1 |
| | IVON-LoRA@mean (2x iteration) | $74.1_{0.6}$ | $76.1_{0.5}$ | $89.5_{0.2}$ | $79.5_{0.3}$ | $83.4_{0.3}$ | $87.7_{0.2}$ | (+2.2) 81.7 |

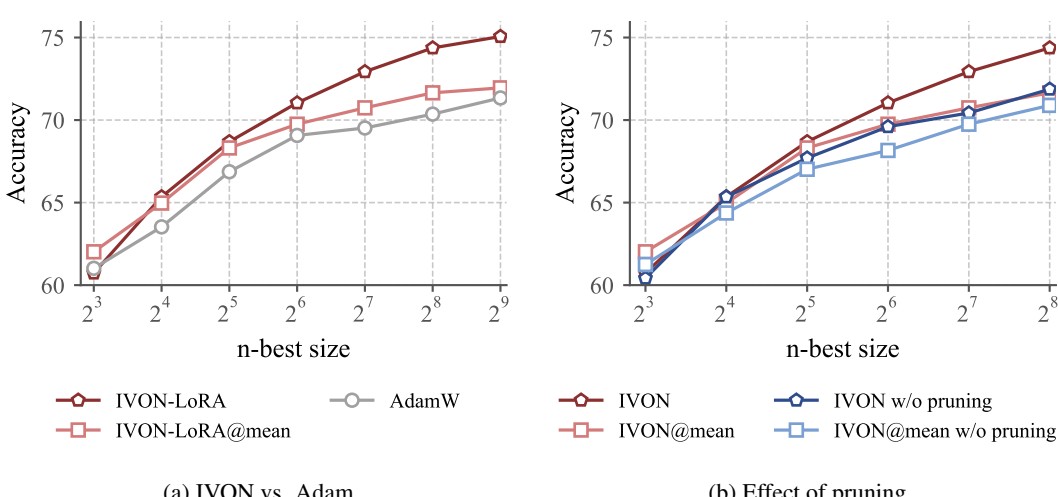

(a) IVON vs. Adam                    (b) Effect of pruning

Figure 3: Improvements obtained with IVON-LoRA on GSM8k increase with n-best-size. For smaller $n$ IVON-LoRA@mean can be most efficient (a). Furthermore, we show that pruning is essential for this, because high-uncertainty parameters are not included when sampling models (b).

multiple outputs for each model sampled from the IVON-LoRA posterior. Then we use a utility function to compare each pair of outputs in the resulting n-best list of size $n$. For GSM8k we use a 0-1-loss and perform majority voting on the final numerical solution. For details on the experimental setup, please refer to App. D.3.

To demonstrate the trend in test-time compute, we scale the size of the n-best list from 8 to 512. For IVON-LoRA@mean and Adam we simply sample the outputs from one model. For IVON-LoRA with posterior we sample 8 outputs each from (n-best-list-size/8) models, i.e. we use 64 model samples from the posterior for $n = 512$ and sample 8 outputs from each model. Specifically, we apply pruning with a ratio of 0.1 for both IVON-LoRA@mean and IVON-LoRA, as we find it consistently improves performance in this setting. Fig. 3a shows that IVON-LoRA@mean outperforms Adam especially for smaller $n$, while as $n$ and the number of models grow the benefit of the posterior become more and more apparent with a 3.7% accuracy improvement of IVON-LoRA for $n = 512$.

In Fig. 3b we further show that pruning greatly benefits both IVON-LoRA@mean and IVON-LoRA. Intuitively, this might be, because sampling high-uncertainty parameters can lead to destructive behavior in the model. Altogether, we show that variational learning combined with uncertainty-aware decoding provides a strong and easy-to-use method for test-time compute scaling.

**Reducing Training Dataset Size.** Finally, we evaluate the performance of IVON-LoRA and Adam when reducing the training dataset size to show the data efficiency of variational learning. We use the same setup as in Sec. 4.1, but randomly subsample 50% of the training data. We show the results in Tab. 4. With training dataset size halved, IVON-LoRA's improvement over Adam is further enlarged from 1.6% to 2.2% on average, indicating that variational learning with IVON-LoRA is more data efficient than Adam.

Table 4: IVON-LoRA achieves even larger improvements over Adam when there is less training data, showing its data efficiency. The experimental setup is the same as in Tab. 1, except that we randomly subsample 50% of the training data. Here we only list accuracy due to space limit, and the full results including ECE and NLL can be found in App. C.1.

| Metrics | Methods | WG-S | ARC-C | ARC-E | WG-M | OBQA | BoolQ | Avg. |
|---------|---------|------|-------|-------|------|------|-------|------|
| ACC ↑ | Adam | $70.5_{0.2}$ | $73.4_{0.7}$ | $88.0_{0.3}$ | $76.8_{0.3}$ | $81.3_{0.6}$ | $86.7_{0.1}$ | 79.5 |
| | IVON-LoRA@mean | $73.7_{0.2}$ | $75.3_{0.3}$ | $88.8_{0.4}$ | $79.0_{0.4}$ | $83.1_{0.4}$ | $86.8_{0.1}$ | (+1.6) 81.1 |
| | Adam (50% training data) | $66.6_{0.6}$ | $70.8_{1.0}$ | $85.2_{0.6}$ | $72.9_{0.7}$ | $79.5_{0.5}$ | $84.5_{0.3}$ | 76.6 |
| | IVON-LoRA@mean (50% training data) | $70.7_{0.6}$ | $73.4_{0.6}$ | $87.6_{0.5}$ | $74.6_{0.5}$ | $81.0_{0.5}$ | $85.1_{0.2}$ | (+2.2) 78.8 |

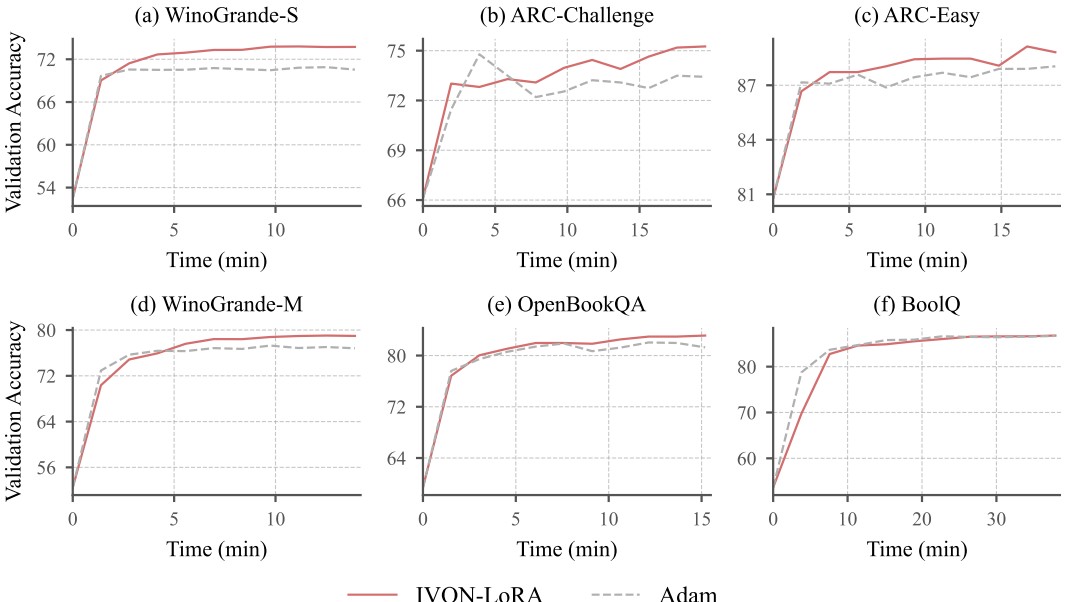

Figure 4: The training speeds of IVON and Adam are similar. We plot validation accuracies (without pruning) of the two methods versus time in minutes. Results are averaged over 5 runs.

To better demonstrate this trend, we also choose the WinoGrande-M dataset (containing 2.6K training examples) and subsample 256, 512, 1024, 2048 training examples to finetune Llama-3.2-3B. We show the results in Fig. 1d. We find that IVON-LoRA consistently outperforms Adam, especially when the size of training dataset is small.

### 4.3 COMPUTATIONAL EFFICIENCY

Finally, we observe that the overhead of IVON-LoRA is negligible. We profile our training code on an NVIDIA GeForce RTX 4090 GPU. In our test run with WinoGrande-S dataset, the forward pass, loss computation, and backward pass of a training step take in total 167.0ms on average. As for the overhead of IVON, the sampling procedure and the optimization step of each training step take 1.0ms and 0.5ms on average, respectively, which is less than 1% of the per-step running time. The overall training speed of IVON-LoRA and Adam are similar as shown in Fig. 4.

## 5 CONCLUSION

In this work, we introduce simple yet effective modifications to variational learning to enhance the LoRA finetuning of LLMs. Our method, IVON-LoRA, adapts the recently proposed IVON optimizer to simplify implementation and reduce overhead associated with Bayesian methods. Furthermore, we propose posterior scaling and pruning techniques that allow for explicit and flexible trade-offs between accuracy and calibration at test time.

Our empirical results demonstrate that IVON-LoRA generally outperforms standard Adam-based LoRA finetuning and other Bayesian methods like Laplace-LoRA and BLoB in terms of both accuracy and calibration, with minimal overhead. We also show that variational learning with IVON-LoRA scales better with increased training and test-time computation, and is more data efficient. Overall, our work presents a practical and effective approach to improve the performance of LoRA finetuning for LLMs.

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

## A    PSEUDOCODE OF IVON-LoRA

We provide the pseudocode of IVON-LoRA in Alg. 1, where modifications to IVON from Shen et al. (2024) are highlighted in red.

---

**Algorithm 1** IVON-LoRA. Modifications to IVON from Shen et al. (2024) are highlighted in red.

---

**Require:** Learning rates $\{\alpha_t\}$, weight-decay $\delta > 0$, scaling factor $\gamma > 0$.
**Require:** Momentum parameters $\beta_1, \beta_2 \in [0, 1)$.
**Require:** Hessian init $h_0 > 0$.
**Init:** $\mathbf{m} \leftarrow$ (LoRA-weights), $\mathbf{h} \leftarrow h_0$, $\mathbf{g} \leftarrow 0$, $\lambda \leftarrow \mathcal{N} \gamma \sqrt{N}$.
**Init:** $\boldsymbol{\sigma} \leftarrow 1/\sqrt{\lambda(\mathbf{h} + \delta)}$.
**Optional:** $\alpha_t \leftarrow (h_0 + \delta)\alpha_t$ for all $t$.
 1: **for** $t = 1, 2, \dots$ **do**
 2:     $\widehat{\mathbf{g}} \leftarrow \widehat{\nabla}\bar{\ell}(\boldsymbol{\theta})$, where $\boldsymbol{\theta} \sim q$
 3:     $\widehat{\mathbf{h}} \leftarrow \widehat{\mathbf{g}} \cdot (\boldsymbol{\theta} - \mathbf{m})/\boldsymbol{\sigma}^2$
 4:     $\mathbf{g} \leftarrow \beta_1 \mathbf{g} + (1 - \beta_1)\widehat{\mathbf{g}}$
 5:     **for** each LoRA matrix index $i$ **do**
 6:         $\boldsymbol{\delta}_i \leftarrow \left( \frac{1}{d_i} \sum_{j=1}^{d_i}(m_{ij}^2 + v_{ij}) \right)^{-1} \cdot \mathbf{1} \in \mathbb{R}^{d_i}$, where $d_i = |\mathbf{m}_i|$
 7:     **end for**
 8:     $\boldsymbol{\delta} \leftarrow \text{concat}(\{\boldsymbol{\delta}_i\})$
 9:     $\mathbf{h} \leftarrow \beta_2 \mathbf{h} + (1 - \beta_2)\widehat{\mathbf{h}} + \frac{1}{2}(1 - \beta_2)^2(\mathbf{h} - \widehat{\mathbf{h}})^2/(\mathbf{h} + \boldsymbol{\delta})$
10:     $\bar{\mathbf{g}} \leftarrow \mathbf{g}/(1 - \beta_1^t)$
11:     $\mathbf{m} \leftarrow \mathbf{m} - \alpha_t(\bar{\mathbf{g}} + \delta \mathbf{m})/(\mathbf{h} + \boldsymbol{\delta})$
12:     $\boldsymbol{\sigma} \leftarrow 1/\sqrt{\lambda(\mathbf{h} + \boldsymbol{\delta})}$
13: **end for**
14: **return** $\mathbf{m}, \boldsymbol{\sigma}$

---

## B    DERIVATION FOR EQ. 2

Here we provide the derivation for Eq. 2 which is based on Graves (2011, Eq. 15), for completeness. For the $i$-th LoRA matrix with prior $p(\boldsymbol{\theta}) = \mathcal{N}(\mathbf{0}, \mathbf{I}/\delta_i)$ and posterior $q(\boldsymbol{\theta}) = \mathcal{N}(\mathbf{m}_i, \text{diag}(\mathbf{v}_i))$, the KL divergence term in Eq. 1 is given by:

$$\mathbb{D}_{\text{KL}}[q(\boldsymbol{\theta}) \,\|\, p(\boldsymbol{\theta})] = \frac{1}{2} \sum_{j=1}^{d_i} \left( \delta_i(m_{ij}^2 + v_{ij}) - 1 - \log \delta_i v_{ij} \right),$$

where $d_i$ is the number of parameters in that matrix and $m_{ij}$ and $v_{ij}$ are the $j$-th entries of $\mathbf{m}_i$ and $\mathbf{v}_i$, respectively. The derivative of the KL divergence with respect to $\delta_i$ is:

$$\frac{\partial}{\partial \delta_i} \mathbb{D}_{\text{KL}}[q(\boldsymbol{\theta}) \,\|\, p(\boldsymbol{\theta})] = \frac{1}{2} \sum_{j=1}^{d_i} \left( m_{ij}^2 + v_{ij} - \frac{1}{\delta_i} \right).$$

Thus we have the stationary point:

$$\delta_i = \left( \frac{1}{d_i} \sum_{j=1}^{d_i}(m_{ij}^2 + v_{ij}) \right)^{-1}.$$

## C    ADDITIONAL RESULTS

### C.1    RESULTS ON THE SCALING OF VARIATIONAL LEARNING

Here we provide the full results of the experiments in Tab. 3 and Tab. 4. Tab. 5 shows the results of training for 10,000 iterations, while Tab. 6 shows the results of using 50% of the training data. Both tables include the ECE and NLL metrics in addition to accuracy.

Table 5: With longer training, IVON-LoRA further improves the performance over Adam. The experimental setup is the same as in Tab. 1, except that we double the number of training iterations for Adam and IVON-LoRA to 10,000. While Adam does not benefit from more training iterations, IVON-LoRA's average accuracy is further improved by 0.6%.

| Metrics | Methods | WG-S | ARC-C | ARC-E | WG-M | OBQA | BoolQ | | Avg. |
|---|---|---|---|---|---|---|---|---|---|
| **ACC** $\uparrow$ | Adam | $70.5_{0.2}$ | $73.4_{0.7}$ | $88.0_{0.3}$ | $76.8_{0.3}$ | $81.3_{0.6}$ | $86.7_{0.1}$ | | 79.5 |
| | Adam (2x iteration) | $71.5_{0.5}$ | $72.6_{0.3}$ | $87.7_{0.4}$ | $77.0_{0.3}$ | $81.8_{0.5}$ | $86.5_{0.1}$ | (+0) | 79.5 |
| | IVON-LoRA@mean | $73.7_{0.2}$ | $75.3_{0.3}$ | $88.8_{0.4}$ | $79.0_{0.4}$ | $83.1_{0.4}$ | $86.8_{0.1}$ | (+1.6) | 81.1 |
| | IVON-LoRA@mean (2x iteration) | $74.1_{0.6}$ | $76.1_{0.5}$ | $89.5_{0.2}$ | $79.5_{0.3}$ | $83.4_{0.3}$ | $87.7_{0.2}$ | (+2.2) | 81.7 |
| **ECE** ($\times100$) $\downarrow$ | Adam | $28.3_{0.2}$ | $24.5_{0.7}$ | $11.0_{0.4}$ | $21.4_{0.3}$ | $16.2_{0.6}$ | $10.1_{0.2}$ | | 18.6 |
| | Adam (2x iteration) | $27.7_{0.7}$ | $25.9_{0.3}$ | $11.4_{0.5}$ | $21.7_{0.4}$ | $16.3_{0.4}$ | $11.8_{0.1}$ | (+0.5) | 19.1 |
| | IVON-LoRA@mean | $21.9_{0.2}$ | $20.2_{0.3}$ | $9.3_{0.4}$ | $16.3_{0.3}$ | $12.6_{0.2}$ | $7.1_{0.1}$ | (-3.9) | 14.7 |
| | IVON-LoRA@mean (2x iteration) | $23.9_{0.5}$ | $22.2_{0.5}$ | $9.5_{0.1}$ | $18.2_{0.4}$ | $14.0_{0.3}$ | $9.3_{0.2}$ | (-2.4) | 16.2 |
| **NLL** $\downarrow$ | Adam | $3.14_{0.10}$ | $2.50_{0.10}$ | $1.01_{0.08}$ | $1.75_{0.08}$ | $1.29_{0.06}$ | $0.57_{0.01}$ | | 1.71 |
| | Adam (2x iteration) | $3.79_{0.33}$ | $3.19_{0.05}$ | $1.20_{0.07}$ | $2.07_{0.10}$ | $1.45_{0.07}$ | $0.89_{0.02}$ | (+0.39) | 2.10 |
| | IVON-LoRA@mean | $1.47_{0.03}$ | $1.41_{0.02}$ | $0.66_{0.02}$ | $0.89_{0.02}$ | $0.79_{0.01}$ | $0.38_{0.00}$ | (-0.78) | 0.93 |
| | IVON-LoRA@mean (2x iteration) | $2.06_{0.05}$ | $1.89_{0.03}$ | $0.85_{0.03}$ | $1.41_{0.02}$ | $1.09_{0.03}$ | $0.55_{0.01}$ | (-0.40) | 1.31 |

Table 6: IVON-LoRA achieves even larger improvements over Adam when there is less training data, showing its data efficiency. The experimental setup is the same as in Tab. 1, except that we randomly subsample 50% of the training data.

| Metrics | Methods | WG-S | ARC-C | ARC-E | WG-M | OBQA | BoolQ | | Avg. |
|---|---|---|---|---|---|---|---|---|---|
| **ACC** $\uparrow$ | Adam | $70.5_{0.2}$ | $73.4_{0.7}$ | $88.0_{0.3}$ | $76.8_{0.3}$ | $81.3_{0.6}$ | $86.7_{0.1}$ | | 79.5 |
| | IVON-LoRA@mean | $73.7_{0.2}$ | $75.3_{0.3}$ | $88.8_{0.4}$ | $79.0_{0.4}$ | $83.1_{0.4}$ | $86.8_{0.1}$ | (+1.6) | 81.1 |
| | Adam (50% training data) | $66.6_{0.6}$ | $70.8_{1.0}$ | $85.2_{0.6}$ | $72.9_{0.7}$ | $79.5_{0.5}$ | $84.5_{0.3}$ | | 76.6 |
| | IVON-LoRA@mean (50% training data) | $70.7_{0.6}$ | $73.4_{0.6}$ | $87.6_{0.5}$ | $74.6_{0.5}$ | $81.0_{0.5}$ | $85.1_{0.2}$ | (+2.2) | 78.8 |
| **ECE** $\downarrow$ | Adam | $28.3_{0.2}$ | $24.5_{0.7}$ | $11.0_{0.4}$ | $21.4_{0.3}$ | $16.2_{0.6}$ | $10.1_{0.2}$ | | 18.6 |
| | IVON-LoRA@mean | $21.9_{0.2}$ | $20.2_{0.3}$ | $9.3_{0.4}$ | $16.3_{0.3}$ | $12.6_{0.2}$ | $7.1_{0.1}$ | (-3.9) | 14.7 |
| | Adam (50% training data) | $32.1_{0.7}$ | $26.9_{1.2}$ | $13.7_{0.4}$ | $25.9_{0.6}$ | $18.2_{0.5}$ | $13.4_{0.2}$ | | 21.7 |
| | IVON-LoRA@mean (50% training data) | $26.2_{0.6}$ | $23.3_{0.8}$ | $10.4_{0.3}$ | $21.8_{0.5}$ | $15.2_{0.6}$ | $10.7_{0.1}$ | (-3.8) | 17.9 |
| **NLL** $\downarrow$ | Adam | $3.14_{0.10}$ | $2.50_{0.10}$ | $1.01_{0.08}$ | $1.75_{0.08}$ | $1.29_{0.06}$ | $0.57_{0.01}$ | | 1.71 |
| | IVON-LoRA@mean | $1.47_{0.03}$ | $1.41_{0.02}$ | $0.66_{0.02}$ | $0.89_{0.02}$ | $0.79_{0.01}$ | $0.38_{0.00}$ | (-0.78) | 0.93 |
| | Adam (50% training data) | $3.90_{0.25}$ | $2.97_{0.27}$ | $1.39_{0.03}$ | $2.75_{0.15}$ | $1.51_{0.10}$ | $0.97_{0.03}$ | | 2.25 |
| | IVON-LoRA@mean (50% training data) | $1.71_{0.08}$ | $1.71_{0.03}$ | $0.76_{0.03}$ | $1.34_{0.04}$ | $0.99_{0.03}$ | $0.60_{0.01}$ | (-1.06) | 1.19 |

Table 7: Performance comparison on the test sets of GLUE benchmark using DeBERTa-v3-base as the base model. Results are averaged over 5 runs with different random seeds.

| Method | CoLA | MRPC | RTE | STS-B | QQP | QNLI | SST-2 | Avg. |
|---|---|---|---|---|---|---|---|---|
| Full-FT | $68.0_{0.5}$ | $90.1_{0.5}$ | $79.4_{1.4}$ | $90.9_{0.1}$ | $90.6_{0.0}$ | $94.0_{0.1}$ | $95.5_{0.1}$ | 86.9 |
| Vanilla LoRA | $68.6_{0.9}$ | $88.9_{0.4}$ | $84.0_{1.1}$ | $90.9_{0.1}$ | $91.1_{0.0}$ | $94.3_{0.1}$ | $95.3_{0.1}$ | 87.6 |
| IVON-LoRA | $68.6_{0.2}$ | $90.6_{0.3}$ | $86.2_{0.5}$ | $91.1_{0.1}$ | $90.7_{0.0}$ | $94.1_{0.1}$ | $95.4_{0.1}$ | 88.1 |

## C.2 Results on GLUE Benchmark

Here we evaluate IVON-LoRA on the GLUE benchmark (Wang et al., 2019). We use DeBERTa-v3-base (He et al., 2021) and compare our method to full-parameter finetuning and LoRA with Adam. Please refer to App. D.4 for details on the experimental setup. We present the results in Tab. 7. Similar to the results in Hu et al. (2022), we observe that LoRA achieves a higher average score than full-parameter finetuning. Notably, IVON-LoRA outperforms vanilla LoRA by 0.5 on average.

## D Details on Experimental Setup

### D.1 General Setup

We utilize the PEFT (Mangrulkar et al., 2022) library for LoRA adaptation, and apply LoRA to the query and value weights of the attention layers. Unlike Yang et al. (2024), we do not apply LoRA to the output layer due to numerical instability encountered in preliminary experiments. For all experiments with LoRA, we set the rank $r$ to 8, $\alpha$ to 16, and LoRA dropout rate to 0.1, which are the default settings in the PEFT (Mangrulkar et al., 2022) library.

### D.2 Commonsense Reasoning

We finetune Llama-3.2-3B (Grattafiori et al., 2024) on six commonsense reasoning datasets, including WinoGrande-S (WG-S), WinoGrande-M (WG-M) (Sakaguchi et al., 2021), ARC-Challenge (ARC-C), ARC-Easy (ARC-E) (Clark et al., 2018), OpenBookQA (OBQA) (Mihaylov et al., 2018), and BoolQ (Clark et al., 2019). For better computational efficiency, the base model undergoes int8 quantization, with LoRA weights maintained in 16-bit precision. Finetuning is performed on a single NVIDIA A100 Tensor Core GPU (with 80 GB memory, for BoolQ dataset due to the memory requirements for longer context) or GeForce RTX 4090 GPU (with 24 GB memory, for other datasets).

To finetune a pretrained language model which predicts the next token in a sequence for solving multiple-choice or true/false questions, we need to wrap the text and the choice of each question into an instruction using predefined prompt templates. We then use the pretrained model to predict the next token of the wrapped instruction, and extract the output logits for the tokens standing for "True"/"False" or "A"/"B"/"C"/"D" choices. For the prompt templates, we use the same ones as in Yang et al. (2024), which are shown in Tab. 8.

Table 8: Prompt templates used for preprocessing different commonsense reasoning datasets.

| Dataset | Prompt Template |
|---|---|
| BoolQ | Answer the question with only True or False: {question} Context: {passage} |
| OBQA / ARC | Select one of the choices that answers the following question: {question} Choices: {choices} Answer: |
| WG | Select one of the choices that answers the following question: {sentence} Choices: A. {option1} B. {option2} Answer: |

### D.2.1 HYPERPARAMETERS

**General Settings**  For all methods, we set the batch size to 8 and the number of training steps to 5,000. We use a linear learning rate scheduler with 150 steps of warmup for Adam and IVON-LoRA and 300 steps for BLoB (which is its default).

**Baseline Methods**  For Adam, we conducted a grid search for the learning rate, and picked $2 \times 10^{-4}$ which performed the best. We set $\beta_1$ to 0.9, $\beta_2$ to 0.999, and $\epsilon$ to $10^{-8}$. For simplicity, we set the weight decay to 0 which is a common practice in LLM finetuning. For Laplace-LoRA and BLoB, we use the same settings as in their code repositories, but we do not truncate the input context and do not finetune the output layer.

**IVON-LoRA**  For IVON-LoRA, we set the learning rate to $1 \times 10^{-2}$, $\beta_1$ to 0.9, $\beta_2$ to 0.9995, and impose an element-wise clipping of preconditioned gradients to 0.005. As shown by Shen et al. (2024), the gradient clipping step is necessary for training language models. We initialize $\mathbf{h}$ to $5 \times 10^{-2}$. For the scaling factor $\gamma$, we set to 5000 for all datasets.

### D.3 GENERATION TASKS

### D.3.1 GSM8K

We finetune Qwen2.5-3B (Team, 2024) on GSM8k (Cobbe et al., 2021) with the hyperparameters described in App. D.3.2. We do not use any quantization and train the model with an effective batch size of 16 on an NVIDIA A100 GPU with 40 GB memory. We use a simple prompt prefix "Solve the following math word problem: " and do not use any in-context exemplars. To perform prediction, we sample a varying number of outputs from the model using ancestral sampling and then perform uncertainty-aware MBR (Daheim et al., 2025). To do so, we extract the final answer following the GSM8k output template and then use a 0-1-loss which reduces to majority voting over the final answer. When using the IVON posterior we always sample 8 outputs from each model. That is, for a total of 32 outputs we use 4 models to sample 8 times and for 64 outputs we sample 8 outputs from 8 models each and so on. Empirically, this has performed better than using fewer models but more outputs in initial experiments.

### D.3.2 HYPERPARAMETERS

**General Settings**  For all methods, we set the batch size to 16. We set the number of training epochs to 6 and use a linear learning rate scheduler.

**Baseline Methods**  For the Adam baseline, similar to the commonsense reasoning tasks, we set the learning rate to $2 \times 10^{-4}$, $\beta_1$ to 0.9, $\beta_2$ to 0.999, $\epsilon$ to $10^{-8}$ and weight decay to 0.

**IVON-LoRA**  For IVON-LoRA, we set the learning rate to $3 \times 10^{-2}$, $\beta_1$ to 0.9, $\beta_2$ to 0.99999, and impose an element-wise clipping of preconditioned gradients to 0.01. Furthermore, we initialize $\mathbf{h}$ to $5 \times 10^{-3}$ and set $\lambda$ to $1 \times 10^7$ without any scaling.

### D.4 GLUE

We finetune DeBERTa-v3-base (He et al., 2021) on the training set of the GLUE benchmark (Wang et al., 2019) separately for each task. Due to its large size and requirements for customized evaluation pipeline, we exclude the MNLI task from our evaluation, which will be included in the final version of the paper. Both the base model and LoRA weights are kept in 16-bit precision. Finetuning is performed on a single GeForce RTX 4090 GPU (with 24 GB memory).

### D.4.1 HYPERPARAMETERS

**General Settings**  Unlike in Hu et al. (2022), we do not conduct extensive hyperparameter search, but instead use a fixed set of hyperparameters for all tasks. We set the number of training epochs for each dataset to the minimum so that the number of epochs is no less than 3 and at least 5000 steps are taken. For all methods, we set the batch size to 16. We use a linear learning rate scheduler

with 6% of warmup steps. Additionally, we truncate the input context to 512 tokens to reduce the memory footprint.

**Baseline Methods** For Adam, we again set $\beta_1$ to 0.9, $\beta_2$ to 0.999, $\epsilon$ to $10^{-8}$ and weight decay to 0. We conduct a grid search for the learning rate for both full-parameter and LoRA finetuning with Adam. For full-parameter and LoRA finetuning, we pick $1 \times 10^{-4}$ and $5 \times 10^{-4}$, respectively.

**IVON-LoRA** For IVON-LoRA, we set the learning rate to $1 \times 10^{-2}$, $\beta_1$ to 0.9, $\beta_2$ to 0.9998, and impose an element-wise clipping of preconditioned gradients to 0.02. We initialize $\mathbf{h}$ to $5 \times 10^{-3}$ and set $\lambda$ to $5 \times 10^6$.

