# OpenReview forum: "Improving LoRA with Variational Learning"
_ICLR.cc/2026/Conference — Submitted to ICLR 2026_

### Official Review · Reviewer_Pxxq · 2025-10-31

**Soundness:** 2
**Presentation:** 3
**Contribution:** 2
**Rating:** 4
**Confidence:** 4

**Summary:**

The paper proposes to improve LoRA via Bayesian machine learning. In particular, the proposal is to integrate the Improved Variational Online Newton method coupled with testing time temperature scaling and posterior pruning. The proposed method also takes into account how to select and fine-tune some hyper-parameters to improve performance. Extensive empirical evaluation shows that the proposed method achieves both high prediction accuracy and well-calibrated models compared to existing methods in the literature.

**Strengths:**

The paper presents an extensive literature review and provide detailed background knowledge about different methods in Bayesian learning for LoRA. The proposed method is also intuitive and easy to follow. The proposed method is efficient in terms of computation. The implementation is also mentioned as minor modification into the existing Adam. The selection of hyper-parameters is justified and has associated with proper references.

Another plus point is the empirical study of tradeoff between accuracy and calibration error. This is known for Bayesian learning, which focuses on modeling uncertainty, and hence, may not be as confident as point estimation. Hence, simply comparing prediction accuracy alone is insufficient.

**Weaknesses:**

The paper is more or less an incremental improvement with certain engineering finetuning. More specifically, it is an improvement of Blob using existing frameworks and hyper-parameter selection to set the prior. It does not mean the paper is bad, but the current contributions are limited.

**Minors**
- Typo at line 158: "... accuracy. using ..." => Using

**Questions:**

Because the paper claims efficiency in terms of computation, it is better to include a complexity analysis in terms of running time for the proposed method and other existing methods to easier compare theoretically.

---

> ### Author Response · Authors · 2025-11-21
>
> We thank the reviewer for the careful reading. Below we address the main concerns.
>
> **W1. On the significance of the contributions**
>
> The reviewer considers our work as "more or less an incremental improvement with certain engineering finetuning" and "an improvement of Blob using existing frameworks and hyper-parameter selection to set the prior". We respectfully but strongly disagree.
>
> Firstly, **our method is built on IVON, not BLoB.** BLoB is built on Bayes by Backprop (BBB, a commonly used variational learning method), with a series of novel techniques to make it practical for LoRA finetuning. IVON-LoRA, as the name suggests, is built on IVON, a new and completely different variational learning method featuring natural gradient updates. Due to the difference in the underlying algorithm, the techniques used in BLoB are not directly applicable to IVON-LoRA, and vice versa.
>
> Next we would like to clarify why we believe the contributions of this work are substantial. The key contributions of the paper are twofold:
>
> 1. A new method, IVON-LoRA, that makes IVON practical for LoRA finetuning of LLMs via three algorithmic adaptations.
> * We use a closed-form empirical-Bayes prior precision for each individual LoRA matrix, based on Eq. (15) of Graves (2011). This removes the need to tune a global prior precision, and it explicitly handles the heterogeneity among different LoRA matrices (different layers and A/B roles), which is a pain point in practice and is discussed in many prior works such as AdaLoRA [1]  and LISA [2]. To our knowledge this combination of IVON-style training and per-matrix empirical Bayes prior has not been explored before.
> * We propose a simple but practically important heuristic which sets $\lambda = \gamma \sqrt{N}$, with a single $\gamma$ shared across datasets of very different sizes. This allows us to use the same $\gamma$ across all commonsense reasoning benchmarks, which avoids per-dataset tuning of $\lambda$.
> * We introduce two test-time techniques for smooth accuracy-calibration trade-off: posterior temperature scaling and SNR-based pruning. They add an easy-to-use calibration knob to IVON-LoRA, which is not available in prior Bayesian LoRA methods.
> 2. Beyond proposing a new method, we also highlight the important trend that variational learning (with IVON-LoRA) scales better than the non-Bayesian baseline when (i) training compute is increased, (ii) test-time compute is increased, and (iii) data is reduced. To our knowledge this scaling behavior has not been reported before in the Bayesian LoRA literature. We highlight such scaling behavior in Fig. 1 (b)-(d) but sadly all four reviewers overlooked this point. We see this as an important trend, because given that compute is growing much faster than high-quality data, methods whose performance improves more with extra compute are especially relevant.
>
> Taken together, our work simplifies IVON for LoRA finetuning to the point where it is just as easy as using Adam, while achieving visibly better accuracy and calibration and revealing an important scaling behavior of variational learning. We hope this clarification will lead the reviewer to reconsider their assessment.
>
> **Q1. On the time complexity analysis of IVON-LoRA and other baselines**
>
> We appreciate the reviewer’s concern on the efficiency of IVON-LoRA and other baselines. We would like to first clarify that algorithmic efficiency is not the only main contribution of this work, as we summarized above.
>
> For Laplace-LoRA, such complexity analysis is beyond the scope of this rebuttal, because it involves techniques such as Low-Rank KFAC and SVD, for which the time complexity is non-trivial and depends on specific choice of algorithms and implementation details.
>
> For IVON-LoRA and BLoB, the theoretical time complexity is the same as Adam and scales linearly with the number of trainable parameters, as they only conduct element-wise operations. However, the constant factors can be very different in practice due to the differences in algorithm design as well as implementation details.

---

> > ### Comment · Reviewer_Pxxq · 2025-11-25
> > **Acknowledgement of the rebuttal**
> >
> > Thank you, the authors, for addressing my concern. Also, thank for clarifying that it is an improved version of IVON, not BLob. Nevertheless, I share the same concern like reviewer Fiag, in which the contribution of this work is still limited as an incremental on previous studies. Beside that, I do not see anything wrong with the paper.

---

### Official Review · Reviewer_LL5V · 2025-10-31

**Soundness:** 3
**Presentation:** 3
**Contribution:** 3
**Rating:** 6
**Confidence:** 3

**Summary:**

The paper investigates Bayesian approaches to Low-Rank Adaptation (LoRA) for fine-tuning Large Language Models. The motivation for such methods is their potential to improve not only model accuracy but also uncertainty calibration. The authors begin by identifying limitations in existing Bayesian LoRA methods, namely Laplace-LoRA and BLoB, arguing that they present a suboptimal trade-off between accuracy and calibration, and can introduce significant computational and implementation complexity. To address these issues, the authors propose a method based on the Improved Variational Online Newton (IVON) algorithm. They introduce several modifications to IVON tailored for LoRA fine-tuning: an online, per-matrix adaptation of the prior precision; a heuristic for setting the scaling parameter of the KL-divergence term in the ELBO; and two test-time techniques, posterior temperature scaling and posterior pruning, to enable an explicit trade-off between accuracy and calibration.

**Strengths:**

- The paper is well-written and the methodological contributions are easy to follow.
- The proposed method, IVON-LoRA, is simple to implement and adds minimal computational overhead. This makes the approach a practical and valuable contribution.
- The empirical evaluation is extensive and robustly demonstrates the benefits of the proposed method through comparisons with:
  - The standard non-Bayesian Adam optimizer, showing that IVON-LoRA improves both accuracy and calibration.
  - Bayesian competitors (Laplace-LoRA and BLoB), showing that IVON-LoRA often achieves a more favorable accuracy-calibration trade-off.

**Weaknesses:**

The motivation for some of the algorithmic design choices could be further substantiated:

- A more in-depth discussion is needed to motivate the adaptation of the prior precision *at each optimization step*. How does this procedure align with the Bayesian framework, where the prior is typically fixed to represent beliefs held before observing the data? Adapting the prior to the current state of the posterior seems to weaken its role as a fixed regularizer, as it no longer serves as a static anchor but rather co-adapts with the posterior.
- The justification for the heuristic choice of the scaling parameter $\lambda$ could be strengthened. The authors argue that a linear scaling, $\lambda = N$, can fail for small $N$ by producing a $\lambda$ that is too small, leading to excessively large posterior variance and training instability. While choosing a larger $\lambda$ is a reasonable response, the specific motivation for a scaled square-root law ($\lambda = \gamma \sqrt{N}$) is not provided. Do the authors have a theoretical or empirical rationale for this functional form over other alternatives? Furthermore, since $\sqrt{N} < N$ for $N>1$, does the success of this heuristic rely on choosing a scaling factor $\gamma \gg \sqrt{N}$ to ensure $\lambda$ is sufficiently large?

I also have some suggestions for improving the experimental evaluation:

- The authors propose posterior pruning and scaling as effective tools for managing the accuracy-calibration trade-off for IVON-LoRA. Could these techniques also be applied to Laplace-LoRA and BLoB? If so, an evaluation of their effectiveness on these baselines would help clarify whether this is a general technique or specific to IVON.
- Similarly, are the reported improvements in scaling with respect to training and test-time computation a general feature of variational learning for LoRA, or are they specific to IVON-LoRA? A small-scale comparison of how BLoB scales with these factors would provide valuable context.
- The paper argues that IVON-LoRA has negligible overhead compared to Adam. However, a direct comparison of computational complexity (e.g., training time, memory usage) against the Bayesian competitors, Laplace-LoRA and BLoB, is missing and would be informative.
- The result tables are dense and can be difficult to parse. To improve readability and facilitate direct comparison, could the authors add the point-estimates for Adam, IVON-LoRA@mean and IVON-LoRA to the accuracy-calibration trade-off plots in Figures 1 and 2?

**Questions:**

cf. my comments below "Weaknesses"

---

> ### Author Response · Authors · 2025-11-21
>
> We thank the reviewer for the careful reading and the positive feedback. Below we address the main concerns.
>
> **W1. On the empirical Bayes prior**
> The main motivation for using an adaptive prior is:
>
> 1. To avoid the need to hand-tune a prior, for which there is no established recipe in the LoRA setting, and the optimal prior can vary across different tasks and datasets.
> 2. To handle the heterogeneity among different LoRA matrices (different layers and A/B roles), rather than using a single global prior precision. Such heterogeneity is a pain point in practice and is discussed in many prior works such as AdaLoRA [1]  and LISA [2]
>
> Having a fixed prior for each matrix might be "more Bayesian", but there is no practical way to find good values for these priors after all, considering the scale of the problem. Thus, we use an empirical Bayes approach to adapt the prior for each matrix.
>
> We agree that more sophisticated updating schemes for the prior are possible, e.g., using a moving average or updating it for every few iterations/epochs. However, we found that the simplest closed-form update at every step already works well in practice. We leave the exploration for better updating schemes for future work.
>
> **W2. On the heuristic choice of $\lambda = \gamma \sqrt{N}$**
>
> Conceptually, $\lambda$ plays the role of an effective dataset size in the generalized posterior. We found empirically that:
>
> * $\lambda$ should monotonically increase with dataset size $N$ to gradually downweight the KL term as more data become available, and
> * As $N$ varies widely across our experiments, a linear scaling $\lambda = \gamma N$ can either produce excessively large posterior variances on small datasets and destabilize training, or too small variances on large datasets, diminishing the benefits of variational learning. Thus, we need a function with a sub-linear growth in $N$.
>
> Considering this, we chose $\sqrt{N}$ as a simple sub-linear function that satisfies these properties and found it works well in practice. Note that other sub-linear functions (e.g., $\log N$) could also be used, but we did not explore them extensively as $\sqrt{N}$ already worked well. The contribution here should be viewed as the identification of the need for sub-linear scaling (which was not discussed in prior IVON literature) and the practical demonstration that $\sqrt{N}$ scaling works well across a range of datasets without per-dataset tuning.
>
> > does the success of this heuristic rely on choosing a very large $\gamma$?
>
> We don't think so, as for a "sufficiently large" $\lambda$, the posterior variance will be very small and it will be equivalent to a point estimation. It cannot explain the gains we observe from Adam baseline to IVON-LoRA, and the differences between IVON-LoRA@mean and IVON-LoRA results.
>
> **W3. On possible improvements to the experimental evaluation**
>
> We thank the reviewer for the detailed suggestions. We added Adam and IVON-LoRA to Fig. 1(a) and Fig. 2 in the latest draft, and are working on adding the other suggested experiments. We will update the draft as soon as the results are ready. Our hypothesis is that, the proposed posterior pruning and scaling techniques should also be applicable to Laplace-LoRA and BLoB, and the scaling advantages should be general to other variational learning methods like BLoB.
>
> As a final note, we are currently working on adding additional experiments to address the concerns raised by all reviewers. We will try to run as many of them as possible during the discussion period, but as some of them require substantial compute, we have to prioritize carefully. It would be very helpful if the reviewer could let us know if the mentioned extra experiments are critical for potentially revising the overall assessment, or nice-to-have experiments for the camera-ready version.
>
> [1] Zhang, Qingru, et al. "Adaptive Budget Allocation for Parameter-Efficient Fine-Tuning." ICLR 2023.
> [2] Pan, Rui, et al. "LISA: Layerwise Importance Sampling for Memory-Efficient Large Language Model Fine-Tuning." NeurIPS 2024\.

---

> ### Author Response · Authors · 2025-11-27
>
> Thank you again for your constructive feedback! Friendly reminder that we have posted a detailed response to your comments to address your concerns, and look forward to discussing this further with you.

---

### Official Review · Reviewer_xHMR · 2025-11-07

**Soundness:** 1
**Presentation:** 3
**Contribution:** 3
**Rating:** 2
**Confidence:** 4

**Summary:**

The paper proposes a variational Bayesian approach, LoRA-IVON, for fine-tuning large language models (LLMs) by adapting a mean-field Gaussian approximation method, IVON, to the LoRA fine-tuning framework. Both IVON and LoRA-IVON are appealing because they can be implemented through minor modifications to the widely used Adam optimizer, making them easily integrable into existing deep learning frameworks. The proposed extensions to IVON include the use of an empirical prior during optimization (i.e., fine-tuning) and the introduction of heuristics for posterior tempering and network pruning to improve posterior predictive sampling. The authors claim that the method achieves superior accuracy and uncertainty calibration with negligible computational overhead, a claim supported by empirical results.

**Strengths:**

- The empirical Bayes formulation for determining the prior is the most novel and promising aspect of the work. This idea has potential applicability beyond the presented context and represents a fundamental improvement over IVON.
- The experimental results provide convincing evidence that the proposed method performs on par with established Bayesian fine-tuning approaches such as BLoB and the Laplace approximation.
- The proposed adaptation preserves the implementation simplicity of IVON.

**Weaknesses:**

- The paper does not include a comparison with an IVON baseline, making it difficult to assess the actual contribution of the proposed adaptations relative to the existing IVON method.

- The choice of tempering with $\lambda$ during optimization is neither theoretically motivated nor empirically investigated. The experiments appear to either fix $\lambda = 5k = \gamma$ or disregard the $\gamma \sqrt{N}$ heuristic entirely (e.g., by setting $\lambda = 5 \cdot 10^6$).

- The calibration analysis should include additional metrics such as the Brier Score and the Maximum Calibration Error to provide stronger evidence for the claimed improvements in calibration. ECE and MCE alone are sensitive to the number of bins.

### Reason for rating
I recommend rejecting the paper in its current form, as it would require considerable rewriting and additional experimentation to convincingly demonstrate empirical improvements over IVON. However, I encourage the authors to perform this evaluation and resubmit, as the approach is promising and the direction worthwhile.

**Questions:**

- Under any reasonable weight-decay scheme, how does IVON compare to the proposed method?
- Please provide the derivation for the closed form of $\delta_i$. In particular, clarify what $\hat{\mu}$ in Eq. 15 of Graves (2011) corresponds to in the equation for $\delta_i$ presented here.
- What is the effect of tuning $\gamma$, and what is the rationale for choosing $\gamma = 5k$?

---

> ### Author Response · Authors · 2025-11-21
>
> We thank the reviewer for the careful reading. Below we address the main concerns.
>
> **W1/Q1. Contribution beyond "plain IVON" and missing IVON baseline.**
>
> The goal of our work is to make variational learning *practically usable* for LoRA finetuning on LLMs, and to show the promising scaling trend of IVON-LoRA, rather than just outperforming a heavily tuned IVON baseline. Running IVON as originally proposed on LoRA is fragile and requires substantial tuning in practice:
>
> * There is no existing reasonable weight-decay / prior precision scheme other than tuning it. Notably, the weight decay term is often set to 0 in LoRA finetuning but this does not fit well with the IVON framework.
> * The KL scaling factor $\lambda$ must be tuned per dataset.
> * No built-in mechanism to control the accuracy-calibration trade-off at test time.
>
> That is to say, to build a decent IVON+LoRA baseline, one must do extensive hyperparameter tuning for each dataset, which is exactly what we want to avoid in practice. In this work, we propose modifications that *remove* most of this tuning burden, but still yield strong empirical performance. We consider this enough to claim a meaningful contribution.
>
> Nevertheless, we agree that a "plain IVON \+ LoRA" baseline is valuable as a reference. We are currently working on this and will update our draft as soon as the results are ready.
>
> **W2/Q3. On the choice of $\lambda$ and the $\lambda = \gamma \sqrt{N}$ heuristic.**
>
> The reviewer mentions that we "fix $\lambda = 5k = \gamma$", which seems to be a misunderstanding. For the commonsense reasoning experiments, we do use the heuristic $\lambda = \gamma \sqrt{N}$ with a fixed $\gamma = 5000$ across all datasets. $\lambda$ varies with the dataset size $N$ and is not fixed to $5000$ or $\gamma$.
>
> For the experiments on generation tasks (GSM8k) we tune $\lambda$ directly, because there is only a single dataset and tuning $\lambda$ is equivalent to tuning $\gamma$.
>
> The GLUE experiments in App. C.2 were run earlier with fixed $\lambda$ of $5 \cdot 10^6$ before we settled on the $\gamma \sqrt{N}$ heuristic. They are meant to be viewed as supplementary experiments and are not used to support the main claims. We believe that with the $\gamma \sqrt{N}$ heuristic, we can achieve better performance on GLUE as well. We are currently running these experiments and will update the draft accordingly once the results are ready.
>
> Still, we agree that the presentation can be improved to avoid confusion, and we will clarify this in the revision.
>
> **Why a $\sqrt{N}$ scaling?** Conceptually, $\lambda$ plays the role of an effective dataset size in the generalized posterior. We found empirically that:
>
> * $\lambda$ should monotonically increase with dataset size $N$ to gradually downweight the KL term as more data become available, and
> * As $N$ varies widely across our experiments, a linear scaling $\lambda = \gamma N$ can either produce excessively large posterior variances on small datasets and destabilize training, or too small variances on large datasets, diminishing the benefits of variational learning. Thus, we need a function with a sub-linear growth in $N$.
>
> Considering this, we chose $\sqrt{N}$ as a simple sub-linear function that satisfies these properties and found it works well in practice. Note that other sub-linear functions (e.g., $\log N$) could also be used, but we did not explore them extensively as $\sqrt{N}$ already worked well. The contribution here should be viewed as the identification of the need for sub-linear scaling (which was not discussed in prior IVON literature) and the practical demonstration that $\sqrt{N}$ scaling works well across a range of datasets without per-dataset tuning.
>
> To answer Q3 specifically, tuning $\gamma$ balances the strength of the KL term in the variational objective. We use a sub-linear scaling with $N$ to ensure that a single $\gamma$ can work well across datasets of varying sizes. $\gamma=5000$ was chosen based on tuning and worked well across all six commonsense datasets without further adjustment. We will clarify this in the revision.

---

> ### Author Response · Authors · 2025-11-21
>
> **W3. On calibration metrics**
>
> In prior literature on Bayesian LoRA (e.g., Laplace-LoRA and BLoB), ECE and NLL are the most commonly used calibration metrics. ECE directly measures calibration error but can be sensitive to binning, NLL captures calibration indirectly but is bin-free. We consider this combination sufficient and follow this convention.
>
> We agree that including additional metrics such as Brier score would strengthen the evidence though. Note that Brier score is not a metric purely for calibration and also captures accuracy. We have added the Brier score to tables 1 and 2 in the latest draft. The conclusions remain unchanged: IVON-LoRA significantly improves Brier score over the Adam baseline and is comparable or outperforms Laplace-LoRA and BLoB as well.
>
> **Q2. Derivation of $\delta\_i$**
>
> We have added the derivation for the adaptive prior precision $\delta\_i$ in App. B. Note that we use a zero-mean Gaussian prior, so there is no need to estimate the $\hat{\mu}$ term in Eq. 15 of Graves (2011).
>
> As a final note, we are currently working on adding additional experiments to address the concerns raised by all reviewers. We will try to run as many of them as possible during the discussion period, but as some of them require substantial compute, we have to prioritize carefully. It would be very helpful if the reviewer could let us know if the mentioned extra experiments are critical for potentially revising the overall assessment, or nice-to-have experiments for the camera-ready version.

---

> ### Comment · Reviewer_xHMR · 2025-11-23
>
> ### W1/Q1
> The critical point here is not that you outperform a heavily tuned IVON+LoRA baseline, but that you provide convincing evidence that *heavy* tuning is necessary for the IVON+LoRA baseline in the first place. This validates the premise of the article and the motivation for your proposed method as it is presented.
>
> “Notably, the weight-decay term is often set to 0 in LoRA fine-tuning, but this does not fit well with the IVON framework.”
>
> I do not follow this point. I would expect that setting weight decay = 0 corresponds to an (improper) uniform prior, which still yields a closed form for $ \mathrm{KL}(N \| U) $. Since IVON requires the gradient of the KL term with respect to $\theta$, the arbitrary scaling constant cancels out, and you can obtain a valid descent direction even with this improper prior.
>
> ---
>
> ### W2/Q3
> I checked, and I did indeed miss that it is $\gamma = 5k$, not $\lambda = \gamma = 5k$, in the reasoning experiment, and that GSM8K is tuned directly. This strengthens the claim, and I have increased the soundness score by one point accordingly.
>
> ---
>
> ### W3
> Thanks for including the Brier score. I have checked, and the Brier scores do follow the same trend as ECE and MCE.
>
> Could you clarify what you mean by “the Brier score is not a metric purely for calibration and also captures accuracy”?
>
> In reliability diagrams, calibration is assessed by comparing accuracy and confidence (e.g., Guo et al. [1], Fig. 1). Likewise, both ECE and MCE are defined in terms of the bin-wise difference between empirical accuracy and predicted confidence (ECE: [1], Eq. 3; MCE: [1], Eq. 5). Since these metrics explicitly depend on empirical accuracy, it is not clear in what sense the Brier score uniquely “captures accuracy” beyond what these calibration metrics already incorporate.
>
> 1. Guo, Chuan, et al. *On calibration of modern neural networks.* ICML, 2017.
>
> ---
>
> ### Q2
> With the derivation in the appendix, Eq. 2 now makes sense. I suggest adding a sentence (perhaps reiterating) that the prior is $N(0, \frac{1}{\delta})$ in the text for Section 3.1 for clarity.
>
> *edit*: fixed prior to variance to precision.

---

### Official Review · Reviewer_Fiag · 2025-11-10

**Soundness:** 2
**Presentation:** 3
**Contribution:** 2
**Rating:** 4
**Confidence:** 3

**Summary:**

The paper proposed to adapt the existing method, IVON, to LoRA for LLMs.
The major contributions are presented in Section 3, which includes:
+ how to find the good values for the prior --- this is achieved by using a closed form expression of $\delta_i$.
+ how to select $\lambda$ --- this is achieved by considering $\lambda = \gamma \sqrt{N}$ and tuning $\gamma$ instead.
+ how to balance the trade-off of accuracy and calibration performance for this kind of Bayesian LoRA --- this is achieved by
    +  taking $\lambda_{test} = \tau \lambda$ and tuning the value of $\tau$. This can scale the posterior variance to balance the trade-off between the accuracy and calibration.
    + pruning a small fraction of parameters at test time using $|m_i|/\sqrt{v_i}$ as a metric.

For the experiments:
   + The author compared the proposed method to existing probabilistic LoRA methods including standard LoRA with Adam, Laplace-LoRA and BLoB on 6 datasets including BoolQ for the model Llama-3.2-3B.
   + However, the authors use average accuracy/ECE on 6 datasets as an overall evaluation metric for these methods, this is a huge mistake. For instance, as shown in Table 2, the proposed method actually have mixed performance on ECE and NLL.

Overall, the contributions are limited, and using the average accuracy/ECE values as evaluation metrics is problematic.

**Strengths:**

+ The paper is clearly written and is easy to follow.
+ Related works are discussed though being limited to the literature of LoRA for LLMs.
+ The contributions of the proposed method are clearly presented and discussed.
+ Despite of the problematic evaluation metric, the authors have conducted good amount of experiments to demonstrate the performance of their method.

**Weaknesses:**

+ The contributions are minor. See Section Summary for a detailed summary of the contributions of this paper.
+ Problematic overall evaluation metric. The authors used average accuracy/ECE on 6 datasets as an overall evaluation metric for these methods, e.g.Table 1 and 2. This is a huge mistake and could cause misleading results and conclusions.
+ Some typos: eg Line 158.

**Questions:**

Q1. The authors mentioned that "we disable the explicit weight decay in the parameter update step, as a
zero weight decay empirically works well for LoRA finetuning" in Line 207-208. Does it because it have similar effect as the KL divergence between $q(\theta)$ and $p(\theta)$ in Eq (1). Have you conduct some ablation studies on this problem as I believe this is a well-known property in the field of variational inference of neural networks.

Q2. To select $\lambda$, the authors proposed to use $\lambda = \gamma \sqrt{N}$ of which $\gamma$ need to be tuned. So why don't you just tune  $\lambda$?

---

> ### Author Response · Authors · 2025-11-21
>
> We thank the reviewer for the careful reading. Below we address their main concerns.
>
> **W1. On the significance of the contributions**
>
> The reviewer considers our contributions “minor”. We respectfully disagree and clarify why we believe they are substantial. The key contributions of the paper are twofold:
>
> 1. A new method, IVON-LoRA, that makes IVON practical for LoRA finetuning of LLMs via three algorithmic adaptations.
> * We use a closed-form empirical-Bayes prior precision for each individual LoRA matrix, based on Eq. (15) of Graves (2011). This removes the need to tune a global prior precision, and explicitly handles the heterogeneity among different LoRA matrices (different layers and A/B roles), which is a pain point in practice and is discussed in many prior works such as AdaLoRA [1]  and LISA [2]. To our knowledge this combination of IVON-style training and per-matrix empirical Bayes prior has not been explored before.
> * We propose a simple but practically important heuristic which sets $\lambda = \gamma \sqrt{N}$, with a single $\gamma$ shared across datasets of very different sizes. This allows us to use the same $\gamma$ across all commonsense reasoning benchmarks and avoid per-dataset tuning of $\lambda$.
> * We introduce two test-time techniques for smooth accuracy-calibration trade-off: posterior temperature scaling and SNR-based pruning. They add an easy-to-use calibration knob to IVON-LoRA, which is not available in prior Bayesian LoRA methods.
> 2. Beyond proposing a new method, we also highlight the important trend that variational learning (with IVON-LoRA) scales better than the non-Bayesian baseline when (i) training compute is increased, (ii) test-time compute is increased, and (iii) data is reduced. To our knowledge this scaling behavior has not been reported before in the Bayesian LoRA literature. We highlight such scaling behavior in Fig. 1 (b)-(d) but sadly all four reviewers overlooked this point. We see this as an important trend, because given that compute is growing much faster than high-quality data, methods whose performance improves more with extra compute are especially relevant.
>
> Taken together, our work simplifies IVON for LoRA finetuning to the point where it is just as easy as using Adam, while achieving visibly better accuracy and calibration and revealing an important scaling behavior of variational learning. We hope this clarification will lead the reviewer to reconsider their assessment.
>
> **W2. On using averaged metrics over datasets**
>
> The reviewer considers the use of average accuracy/ECE across six datasets a “huge mistake” that could lead to misleading conclusions. We would like to clarify that:
>
> 1. We do not hide per-dataset results, which are fully reported in Tables 1, 2 and Fig. 2\. The averaged scores are simply a concise summary, not the sole evidence of our claims.
> 2. Aggregated scores are standard metrics in widely used multi-task benchmarks such as GLUE and MMLU. Our use of the average is in line with this convention and is not intended to obscure heterogeneous behavior.
>
> Nevertheless, we will improve the text to make the connection between per-dataset and averaged results more explicit.
>
> **W3. Typos**
> We thank the reviewer for pointing out the typo at Line 158\. We have fixed it in the latest version.
>
> **Q1. On disabling explicit weight decay**
>
> We thank the reviewer for raising this interesting question. We disable explicit weight decay mainly because no weight decay is the default and arguably simplest choice for LoRA finetuning. The Adam baseline also uses no weight decay.
>
> Nevertheless, we agree that whether adding explicit weight decay would help is an interesting question. Reaching a definitive conclusion would require a series of tuning experiments for IVON-LoRA, which we are working on and will report when ready.
>
> **Q2. On tuning $\gamma$ instead of $\lambda$**
>
> In our setting, the dataset size $N$ varies substantially across datasets. We would like a single hyperparameter that works across datasets, rather than tuning $\lambda$ for each one. By writing $\lambda$ as $\lambda = \gamma \sqrt{N}$ we only need to tune the scale $\gamma$ once on a representative dataset and then reuse it for all datasets.
>
> As a final note, we are currently working on adding additional experiments to address the concerns raised by all reviewers. We will try to run as many of them as possible during the discussion period, but as some of them require substantial compute, we have to prioritize carefully. It would be very helpful if the reviewer could let us know if the mentioned extra experiment is critical for potentially revising the overall assessment, or a nice-to-have experiment for the camera-ready version.
>
> [1] Zhang, Qingru, et al. "Adaptive Budget Allocation for Parameter-Efficient Fine-Tuning." ICLR 2023.
> [2] Pan, Rui, et al. "LISA: Layerwise Importance Sampling for Memory-Efficient Large Language Model Fine-Tuning." NeurIPS 2024.

---

> > ### Comment · Reviewer_Fiag · 2025-11-25
> >
> > Thanks for the comments.
> >
> > Your statements in '1. A new method, ...' is a summary of your method and contributions but it doesnot change the problem of incremental contribution.
> >
> > I tend to maintain my current score. I will wait until later to finalize my assessment and discuss with other reviewers and AC.

---

### Author Response · Authors · 2025-12-03

Dear AC,

Thank you for taking over this paper. We appreciate your time and effort, especially given the current difficult situation.

We would like to briefly explain why we believe the current overall assessment is too negative (especially the reviews by Reviewer Fiag and Pxxq who describe the contributions as "minor" or "incremental" and regard it as the *only main reason* for their low scores), and kindly ask you to reconsider the evaluation regarding the paper's broader significance. Our central goal is to *make variational learning practical and effective for LoRA finetuning of LLMs*, which is achieved through three adaptations of IVON: (1) an empirical-Bayes, per-matrix prior precision that removes a difficult global hyperparameter and handles heterogeneity between different LoRA matrices; (2) a practical heuristic for $\lambda$ of the form $\lambda = \gamma \sqrt{N}$, which allows users to tune a single $\gamma$ and obtain sensible $\lambda$ values across datasets of very different sizes; and (3) test-time posterior temperature scaling and SNR-based pruning to explicitly control the accuracy–calibration trade-off. These changes significantly lower the barrier to using IVON for LoRA while still yielding clear empirical gains over non-Bayesian LoRA and Bayesian baselines.

A key point that seems to have been neglected in *all* the reviews is the compute-scaling and data-efficiency behavior of variational learning in the LLM LoRA setting. Our experiments in Sec. 4.2 and Fig. 1 show, for the first time to our knowledge, that a variational Bayesian method (IVON-LoRA) scales better than standard Adam-based LoRA when we increase training iterations, increase test-time compute (via posterior sampling with uncertainty-aware MBR decoding), and reduce the amount of training data. In other words, when additional compute is available but data is limited, which is precisely the regime we expect to see more and more often, variational learning yields larger improvements over the non-Bayesian baseline. We view this as a significant finding as important as the method itself, and we would be very grateful if you could consider this point in your re-evaluation.

For Reviewer xHMR, we would like to highlight that their opinion has already shifted positively during the discussion phase: they increased Soundness from 1 to 2, while Presentation and Contribution remain at 3, which does not seem compatible with an overall score of 2. Their main remaining concern is that a "plain IVON" baseline might require substantial tuning to work well; we fully agree this is an important comparison, but it is also technically non-trivial, since the prior precision and $\lambda$ terms are in the variational objective which IVON optimizes, and are very sensitive in a dataset and layer dependent way. In practice, either one under-tunes IVON and obtains an artificially weak baseline, or one invests a heavy per-dataset tuning effort that is precisely what our empirical-Bayes prior and $\lambda$-heuristic are designed to avoid, so the baseline would no longer be comparable to our "practical recipe" for users.

---

### Meta-Review · Area_Chair_oPNQ · 2026-01-02

**Summary:**

The paper proposes various adaptation of the IVON method to make it more suitable for LoRA finetuning. The paper presents interesting findings, for instance regarding scaling behavior (when changing training iterations, inference-time computer, and data availability) for the proposed method. However, the reviewers also raise several issues with the current presentation.

One of the key motivations of the methos is that the proposed adaptations are necessary to make IVON applicable to LoRA finetuning, but the paper does not explicitly demonstrate this claim. Even if the purpose is not to outperform IVON, some form of comparison to show the merits of the proposed method would be necessary, considering that it builds heavily on IVON. This could for instance be accomplished by setting up an numerical evaluation focusing on (lack of) robustness to hyperparameters and showing how the proposed adjustments would make IVON-LoRA more robust than IVON.

Furthermore, the proposed adaptations are largely heuristic and the rationale for these algorithmic design choices could be improved.

Regarding the aforementioned scaling behavior; this is presented for IVON-LoRA but it is not clear if this is a property of this particular method or Bayesian finetuning in general. In case of the latter (which the authors hint at in their rebuttal) this seems to be a different and largely unrelated contribution than the method development.

**Reviewer Concerns:**

Some misunderstandings were sorted out, but comparison with IVON - or more convincing arguments why it's not applicable - to motivate the proposed method development is lacking.

**Reviewer Scores:**

I don't think that the authors have really addressed the main points, so I wouldn't have expected any major changes.

---

### Decision · Program_Chairs · 2026-01-26

Reject